# Decoding TRIP13’s Role in Gastric Cancer: Implications for Prognosis and Immune Response

**DOI:** 10.3390/biomedicines13092268

**Published:** 2025-09-15

**Authors:** Tongguo Shi, Yu Shen, Anjing Zhao, Rufang Dong, Fan Chen, Suhua Xia

**Affiliations:** 1Jiangsu Institute of Clinical Immunology, The First Affiliated Hospital of Soochow University, Suzhou 215000, China; shitg@suda.edu.cn (T.S.); shenyu023015@163.com (Y.S.);; 2Department of Oncology, The First Affiliated Hospital of Naval Military Medical University, Shanghai 200433, China; dr_zhaoaj@163.com; 3Department of Oncology, The First Affiliated Hospital of Soochow University, Suzhou 215000, China

**Keywords:** gastric cancer, *TRIP13*, prognosis, ferroptosis, immune

## Abstract

**Background/Objectives:** Gastric cancer (GC), a prevalent global malignancy and a leading cause of cancer-related mortality, has a poorly understood prognosis related to *TRIP13* expression. *TRIP13* has a recognized part in driving tumor progression across different cancer types, yet its precise role in GC remains beyond our full comprehension. Our study aimed to explore *TRIP13’s* prognostic value and function in GC patients. **Methods**: We extensively explored *TRIP13’s* influence on GC prognosis, functionality, and immune response by examining various cancer-related databases like UALCAN, GEPIA, GEO, and TIMER. Immunohistochemistry (IHC) staining was also conducted to assess the link between *TRIM13* and GC patient survival. **Results**: *TRIP13* expression levels were found to be significantly elevated in GC tissues compared to normal tissues through analysis of mRNA data from multiple public databases. IHC analysis exposed elevated TRIP13 protein levels in GC tissues and connected it with tumor depth. Prognostic evaluation demonstrated that GC patients exhibiting heightened TRIP13 expression endured a diminished overall survival rate. Gene Ontology (GO) and Kyoto Encyclopedia of Genes and Genomes (KEGG) analyses showed that genes related to *TRIP13* are involved in processes such as the cell cycle and DNA repair. Additionally, *TRIP13* expression was found to correlate with ferroptosis-related genes and may play a role in regulating ferroptosis. Immune cell infiltration analysis demonstrated that *TRIP13* expression is negatively correlated with the infiltration of CD4^+^ T cells, CD8^+^ T cells, and B cells. **Conclusions**: *TRIP13* emerges as a candidate independent prognostic indicator and a promising intervention point for GC treatment.

## 1. Introduction

Gastric cancer (GC) is a widely occurring malignant tumor globally, standing fifth in incidence rate and cancer-related mortality [1]. It is predominantly found in East Asia, Eastern Europe, and South America, with a higher incidence in males compared to females [2]. Various factors contribute to the risk of GC, including *Helicobacter pylori* (*H*. *pylori*) infection, Epstein–Barr virus (EBV) infection, alcohol consumption, smoking, aging, high-salt and low-fiber diets, and family factors [3]. A significant proportion of GC cases are diagnosed at a late stage owing to the delayed presentation of clinical symptoms. This factor enhances the invasive potential of GC cells and renders the disease a severe and often fatal threat to human health [4]. Therefore, the urgent need to identify therapeutic targets for GC has now become a critical imperative in oncological research.

Thyroid hormone receptor-interacting protein 13 (*TRIP13*), an AAA^+^ ATPase superfamily protein, orchestrates DNA spindle assembly checkpoints and DNA repair pathways during mitosis [5]. Growing evidence indicates that *TRIP13* is overexpressed in a spectrum of malignancies, such as those affecting the lung, head and neck, breast, colorectum, liver, and prostate [5]. It performs pivotal functions in key tumor progression activities, including cell proliferation, invasive spread, migratory movement, and distant metastasis. For example, within the context of GC, *TRIP13* drives malignant cell proliferation, migration, and invasion through the stabilization of DExD-box helicase 21 (*DDX21*) expression levels [6]. Similarly, in non-small cell lung cancer, the overabundance of *TRIP13* intensifies resistance to gefitinib through the manipulation of autophagy processes and the alteration of phosphorylation within the epidermal growth factor receptor (EGFR) signaling pathway [7]. Notwithstanding these findings, there is a requirement for more extensive and comprehensive research to ascertain whether *TRIP13* possesses the necessary attributes to serve as a promising biomarker for GC. 

In this study, we sought to evaluate the prognostic significance of *TRIP13* in GC. Utilizing a comprehensive bioinformatics approach across multiple public mRNA databases, we identified a marked upregulation of *TRIP13* expression in GC tissues relative to normal tissues. Further analysis revealed that elevated *TRIP13* protein levels in GC tissues are correlated with tumor depth. Importantly, GC patients with high *TRIP13* expression experienced a reduced overall survival rate. Moreover, *TRIP13* expression levels were found to be inversely associated with the infiltration of CD4^+^ T cells, CD8^+^ T cells, and B cells.

## 2. Materials and Methods

### 2.1. Analysis of TRIP13 mRNA Expression

Pan-cancer analysis of *TRIP13* mRNA expression was performed across tumor and normal tissues using the UALCAN tool (https://ualcan.path.uab.edu/, accessed on 21 April 2025). Additionally, the expression of *TRIP13* mRNA in GC and normal tissues was analyzed via GEPIA2 (http://gepia2.cancer-pku.cn/#index, accessed on 27 April 2025), UALCAN, and ENCORI (https://rnasysu.com/encori/, accessed on 6 June 2025) tools. Moreover, we utilized GSE54129 (21 normal tissues and 111 GC tissues), GSE27342 (80 normal tissues and 75 GC tissues), and GSE56807 (5 normal tissues and 5 GC tissues) to analyze *TRIP13* mRNA expression levels in normal and GC tissues.

### 2.2. GC Specimens and Immunohistochemistry (IHC) Staining

This investigation utilized a GC tissue microarray (TMA) encompassing 81 GC specimens along with 81 corresponding normal tissues, all accompanied by follow-up information, which was furnished by Shanghai Outdo Biotech Co., Ltd (Shanghai, China). The attributes of the patients are delineated in Table 1. The ethical endorsement for the study was bestowed by the Medical Ethics Committee of the First Affiliated Hospital of Soochow University (Ethics No. 2023058). The IHC staining procedure is described below: Following antigen retrieval, the tissue sections were cultivated with a human *TRIP13* antibody (Proteintech) overnight at 4 °C, then exposed to a horseradish peroxidase-linked secondary antibody for 1 h at ambient temperature. 3,3′-diaminobenzidine (DAB) was utilized for imaging. The IHC score was determined by multiplying the staining intensity (absent: 0; slight: 1; intermediate: 2; pronounced: 3) by the staining area (≤25%: 1; 25–50%: 2; 50–75%: 3; >75%: 4) [8,9].

### 2.3. Survival Prognosis Assessment

To assess the relationship between *TRIP13* mRNA levels and overall survival, we utilized the Kaplan-Meier Plotter database (https://kmplot.com/analysis/, accessed on 17 June 2025) by constructing forest plots superimposed on Kaplan–Meier curves. Patients were stratified into low and high expression groups using the auto-select best cutoff function in the Kaplan–Meier Plotter database.

In our cohort, individuals with GC were allocated to two distinct groups according to *TRIP13* expression levels: a low-expression cohort (IHC score ≤ 6, *n* = 46) and a high-expression cohort (IHC score > 6, *n* = 45). The Kaplan-Meier methodology was employed to conduct the overall survival analysis, and the log-rank test was used to assess the statistical significance of the survival data. Further, the Cox proportional hazards model analysis was executed using R version 3.5.3.

### 2.4. Function and Pathway Analysis

We acquired STAR-counts data and clinical details for GC from the TCGA database through the GDC Data Portal (https://portal.gdc.cancer.gov, accessed on 7 July 2025). The data were extracted in TPM format and normalized using log2(TPM+1) transformation. After retaining samples with both RNA-seq and clinical data, we selected *TRIP13* high expression GC samples (*n* = 188) and low expression samples (*n* = 187) for analysis.

To study mRNA differential expression, we used the R Limma package (version 3.40.2). In the TCGA dataset, adjusted *p*-values were scrutinized to account for false positives. The criterion for significant differential expression was established as adjusted *p* < 0.05 and |log2(fold change)| > 2.

We used the R ClusterProfiler package to perform functional enrichment analysis. Gene Ontology (GO) analysis (focusing on molecular function, biological process, and cellular components) and Kyoto Encyclopedia of Genes and Genomes (KEGG) pathway enrichment analysis were performed to explore the functions and pathways of the target genes [10].

For the single-sample gene set enrichment analysis (ssGSEA), we compiled gene sets from the relevant pathways and analyzed them using the GSVA package in R software v4.0.3 with the parameter method set to “ssgsea.” Spearman correlation analysis was then applied to examine the correlation between gene expression and pathway scores [11,12,13]. All statistical evaluations were executed within the R programming environment (version v4.0.3), with outcomes having a *p*-value below 0.05 being taken as evidence of statistical significance.

To analyze the relationship between *TRIP13* and ferroptosis, ferroptosis-associated genes were pinpointed through Ze-Xian Liu et al.’s comprehensive exploration of ferroptosis irregularities and roles in cancer (Appendix A) [14,15]. Statistical analysis was performed using R software (version v4.0.3), and results were considered statistically significant at a *p*-value threshold of less than 0.05.

### 2.5. Immune Correlation Analysis

To analyze GC data, we acquired STAR-counts and clinical information from the TCGA database (https://portal.gdc.cancer.gov, accessed on 28 July 2025). The data were obtained in TPM format and underwent normalization through log2(TPM+1) conversion. Upon screening for specimens that possessed both RNA-seq and clinical information, a total of 375 GC samples were chosen for further analysis.

For immune score assessment, we used the R package (v 4.1.3) immunedeconv, which integrates six advanced algorithms (TIMER, EPIC, quanTIseq, and CIBERSORT). These algorithms were chosen for their unique strengths and validated performance. Results were analyzed and visualized using the R package (v 4.1.3) ggClusterNet [16]. All analyses were performed using R version v4.1.3, with *p* < 0.05 considered statistically significant.

Moreover, Spearman’s correlation analysis was executed to evaluate the correlation between *TRIP13* and immune cells (CD4^+^ T cells, CD8^+^ T cells, B cells, neutrophil, macrophage and myeloid dendritic cells) in GC using TCGA data (*n* = 375) [17,18,19]. Statistical analysis was conducted using R software, version v4.0.3. Results were considered statistically significant when the *p*-value was less than 0.05.

### 2.6. Statistical Analysis

GraphPad Prism 8 (GraphPad Software, San Diego, CA, USA) were used for both data analysis and visualization. The *p*-value < 0.05 was considered statistically significant.

## 3. Results

### 3.1. Transcriptional Levels of TRIP13 in GC Tissues

We initially examined *TRIP13* mRNA expression in various human cancers and their corresponding normal samples from the TCGA database via the UALCAN website (Figure 1A). *TRIP13* exhibited elevated expression in numerous malignancies, including bladder cancer (BLCA), breast cancer (BRCA), cervical squamous cell carcinoma (CESC), colon adenocarcinoma (COAD), esophageal carcinoma (ESCA), glioblastoma multiforme (GBM), head and neck squamous cell carcinoma (HNSC), kidney renal papillary cell carcinoma (KIRP), kidney renal clear cell carcinoma (KIRC), liver hepatocellular carcinoma (LIHC), lung adenocarcinoma (LUAD), pancreatic adenocarcinoma (PAAD), lung squamous cell carcinoma (LUSC), prostate adenocarcinoma (PRAD), rectum adenocarcinoma (READ), sarcoma SARC), stomach adenocarcinoma (STAD), and uterine corpus endometrial carcinoma (UCEC) (*p* < 0.05). However, *TRIP13* mRNA expression showed no significant changes in kidney chromophobe (KICH), pheochromocytoma and paraganglioma (PCPG), skin cutaneous melanoma (SKCM), thyroid carcinoma (THCA), and thymoma (THYM). Furthermore, based on UALCAN, GEPIA 2.0, and ENCORI databases, *TRIP13* mRNA levels were higher in GC tissues than in normal tissues (Figure 1B–D). Additionally, compared to normal controls (G1), *TRIP13* mRNA expression was significantly upregulated in GC tissues (G2) from GSE54129, GSE27342, and GSE56807 databases (Figure 1E–G).

### 3.2. Elevated TRIP13 Expression Correlates with Clinical Pathological Characteristics and Survival in GC Patients

To probe into the protein expression of TRIP13 in GC, an IHC assay was conducted on a GC tissue microarray. As depicted in Figure 2A, representative IHC images revealed higher TRIP13 protein expression in GC tissues compared to non-cancerous adjacent tissues (NAT).

Subsequently, the analysis of the correlation between TRIP13 protein levels and the clinical and pathological features of GC patients was carried out. The findings indicated that *TRIP13* levels were significantly linked to tumor depth (*p* = 0.0253), with no notable associations observed for other clinical features (Table 1).

To evaluate the influence of *TRIP13* expression on the survival of GC patients, patients were categorized into high- and low-expression cohorts in accordance with *TRIP13* levels. Figure 2B illustrates that GC exhibiting elevated *TRIP13* levels demonstrated poorer overall survival compared to those with diminished *TRIP13* expression. Kaplan-Meier plotter analysis further demonstrated that high *TRIP13* expression was linked to unfavorable prognoses in GC patients (Figure 2C). Moreover, Cox regression analysis was leveraged to recognize independent prognostic risk factors influencing overall survival. The results highlighted that lymph node (LN) involvement and *TRIP13* expression were significant risk factors for GC patient survival (Figure 2D). Collectively, these data suggest that upregulated *TRIP13* is connected to GC development and is a marker for poor patient outcomes.

### 3.3. The Potential Biological Role of TRIP13 in GC

To probe into *TRIP13*’s role in GC, we identified differentially expressed genes (DEGs) between GC tissues exhibiting elevated *TRIP13* expression and those with diminished expression, using a threshold of fold change > 2 and *p* < 0.05. A total of 133 upregulated genes and 79 downregulated genes were found in the high *TRIP13* expression GC tissues compared to the low expression ones (Figure 3A,B). Subsequently, GO function and KEGG pathway analyses were performed based on these DEGs.

The GO enrichment analysis indicated that the upregulated genes were mainly associated with mitotic nuclear division, microtubule binding, and spindle (Figure 3C). The KEGG pathway analysis revealed that upregulated factors contributed to cell cycle mechanisms, oocyte meiosis, and the p53 signaling network (Figure 3D). As for the underexpressed genes, they were largely associated with muscle contraction, glycosaminoglycan binding, and collagen-containing extracellular matrix (Figure 3E). In terms of pathways, the downregulated genes were accumulated in the cGMP-PKG signaling pathway, vascular smooth muscle contraction pathway, and calcium signaling pathway (Figure 3F). 

Utilizing the ssGSEA algorithm, we worked out the enrichment rates of defined pathways for each sample to probe into the sample-pathway relationship. The resulting fractions provided an intuitive reflection of the interactions between samples and pathways. Our results indicate that in GC, *TRIP13* may be implicated in multiple key processes, including the tumor proliferation signature, G2M checkpoint regulation, MYC-related pathways, DNA replication, DNA repair mechanisms, apoptosis, pyruvate metabolism, and steroid biosynthesis pathways, as well as cysteine and methionine metabolic processes (Figure 4). Overall, our findings reveal a significant link between *TRIP13* expression levels and the proliferation and metabolic activities of cancer cells in GC, reinforcing its possible importance as a critical factor in gastric carcinogenesis and tumor progression.

### 3.4. Relationship Between TRIP13 Expression and Ferroptosis

Given the established involvement of *TRIP13* expression in metabolic processes (Figure 4), we explored its potential association with ferroptosis in GC. Our study assessed the association between *TRIP13* expression and ferroptosis-related genes in GC tissues. As illustrated in Figure 5, the expression levels of ferroptosis-related genes, such as *CISD1*, *EMC2*, *FANCD2*, *HSPB1*, *NFE2L2*, *SLC1A5*, *FDFT1*, *HSPA5*, *SLC7A11*, *CARS1*, *CS*, *DPP4*, *ACSL4*, *ALOX15*, *ATP5MC3*, *GLS2*, *RPL8*, and *TFRC*, were significantly higher in GC tissues with high *TRIP13* expression than in those with low *TRIP13* expression. Conversely, *ATL1* expression was lower in high *TRIP13* expression GC tissues. However, no significant difference was observed in the expression of *CDKN1A*, *GPX4*, *MT1G*, *SAT1*, *LPCAT3*, and *NCOA4* between high and low *TRIP13* expression GC tissues. These outcomes suggest that *TRIP13* may be a critical component in regulating ferroptosis in GC.

### 3.5. The Association Between TRIP13 Expression and Infiltrating Immune Cells

To examine the relationship between *TRIP13* expression and immune cell infiltration in GC, we conducted a correlation analysis. Network connection diagrams and heatmaps were used to visualize the association between *TRIP13* expression and immune scores, where the intensity of red/blue color and the size of rings indicated the strength of correlation. Red lines denoted negative correlations, while green lines represented positive correlations. Using TIMER, EPIC, QUANTISEQ, and CIBERSORT algorithms, we observed that *TRIP13* expression exhibited a negative association with CD4^+^ T cells, CD8^+^ T cells, and B cells (Figure 6). Spearman correlation analysis on TCGA GC samples via the TIMER database further confirmed this negative correlation. Specifically, *TRIP13* expression showed negative associations with B cells, CD4^+^ T cells, CD8^+^ T cells, macrophages, neutrophils, and myeloid dendritic cells (Figure 7).

## 4. Discussion

As documented in prior studies, *TRIP13* expression is a significant factor in various human cancers [20]. For instance, *TRIP13* contributes to GC cell growth, movement, and penetration in a laboratory setting, along with tumor development and spread in a living organism by stabilizing *DDX21* expression [6]. Sun et al. demonstrated that tetrahydrocurcumin targets *TRIP13*, inhibits TRIP13/USP7/c-FLIP interaction, and mediates c-FLIP ubiquitination in triple-negative breast cancer [21]. In this study, we carried out a thorough analysis of *TRIP13* expression and its roles in GC using multiple public databases.

*TRIP13* is upregulated across diverse oncological categories, such as colorectal [22], lung [23], head and neck [24], breast [25], liver [26], and prostate [27] cancers. Liu et al. found it markedly increased in cervical cancer tissues, with high expression indicating unfavorable prognosis in individuals with cervical cancer [28]. *TRIP13* expression also rises from colorectal adenoma to carcinoma, and its overexpression in colorectal cancers is independent of multiple patient factors [29]. In our pan-cancer analysis, *TRIP13* was significantly upregulated in multiple tumors. Through UALCAN, GEPIA 2.0, ENCORI, GSE54129, GSE27342, and GSE56807 databases, we observed markedly increased *TRIP13* mRNA expression in GC tissues. Similarly, our GC cohort showed upregulated *TRIP13* protein expression in GC tissues, linked to tumor depth. However, our analysis revealed no significant difference in *TRIP13* expression levels between gastric cancer patients with and without *H. pylori* infection using the UALCAN database (Appendix A). Notably, high levels of *TRIP13* mRNA and protein in GC served as a predictor of poor prognosis. Cox regression analysis highlighted lymph node involvement and *TRIP13* expression as key risk factors for GC patient survival. Thus, *TRIP13* upregulation shows a connection to poor prognoses and might be a promising biomarker for GC outcome prediction.

Previous studies have established that *TRIP13* promotes tumor cell proliferation, metastasis, drug resistance, and glycolysis in various cancers [30,31,32]. For instance, in thyroid cancer, *TRIP13* interference suppressed proliferation and metastasis by regulating the TTC5/p53 signaling cascade and genes associated with epithelial-mesenchymal transition [30]. In colorectal cancer, high *TRIP13* expression activated glycolysis, enhancing cell stemness and doxorubicin resistance [32]. Herein, GO and KEGG pathway analyses were conducted to explore *TRIP13*’s role in GC using DEGs between high- and low-*TRIP13* expression GC tissues. The GO analysis revealed that *TRIP13*-related genes were mainly involved in mitotic nuclear division, microtubule binding, spindle, muscle contraction, glycosaminoglycan binding, and collagen-containing extracellular matrix. KEGG pathway analysis showed that these genes participated in the cell cycle, oocyte meiosis, the p53 signaling pathway, the vascular smooth muscle contraction mechanism, the cGMP-PKG signaling cascade, and the calcium signaling pathway. The ssGSEA algorithm results indicated that *TRIP13* might be implicated in multiple key processes, including tumor proliferation signature, G2M checkpoint regulation, MYC-related pathways, DNA replication, DNA repair mechanisms, apoptosis, pyruvate metabolism, steroid biosynthesis pathways, and cysteine and methionine metabolic processes. Overall, these findings imply that *TRIP13* is vital for GC cell proliferation, metastasis, drug resistance, and metabolism.

Ferroptosis, an iron-dependent cell death modality, has emerged as a pivotal process in diverse biological functions, particularly in iron, lipid, and amino acid metabolism [33,34,35]. This has drawn significant attention to its therapeutic potential in GC, providing an innovative method to tackle drug resistance [33,34,35]. For example, cysteine protease inhibitor SN (CST1) has been found to block ferroptosis and promote GC metastasis by stabilizing GPX4 protein via OTUB1 [36]. Additionally, targeting the STAT3-ferroptosis axis has proven effective in curbing tumor growth and reducing chemoresistance in GC [37]. In our study, we found that several ferroptosis-related genes were upregulated in GC tissues with high *TRIP13* expression. This evidence points to *TRIP13* as a pivotal governor of ferroptosis in GC, highlighting its suitability as a therapeutic goal.

Prior studies indicate a close link between *TRIP13* and immune responses in cancers [5]. In hepatocellular carcinoma, high *TRIP13* expression correlates with increased Th2 cell infiltration and reduced infiltration of neutrophils, Th17 cells, and dendritic cells [38]. In endometrial carcinoma, *TRIP13* points to a positive connection with immunosuppressive cell infiltration and a negative connection with immune-activating cell infiltration [39]. Pan-cancer analysis also shows *TRIP13* expression is related to immunocyte infiltration and immune scores in specific cancers [40]. Herein, we explore the *TRIP13*-immune cell infiltration relationship in GC. Network diagrams and heatmaps present a negative correlation between *TRIP13* expression and CD4^+^ T cells, CD8^+^ T cells, and B cells. Spearman correlation analysis via the TIMER database confirms this. Thus, *TRIP13* may serve as an immunotherapy target or biomarker for GC and could indicate immunotherapy effectiveness.

## 5. Conclusions

*TRIP13* may function as an independent prognostic biomarker in GC patients. Our study’s findings, derived from a comprehensive online database and validated via clinical samples, indicate this. Yet, more clinical data are needed for further validation. Additionally, in-depth studies are necessary to clarify *TRIP13*’s clinical role and immune escape mechanism in GC. These results offer a valuable reference for future research on *TRIP13*’s role in GC.

## Figures and Tables

**Figure 1 biomedicines-13-02268-f001:**
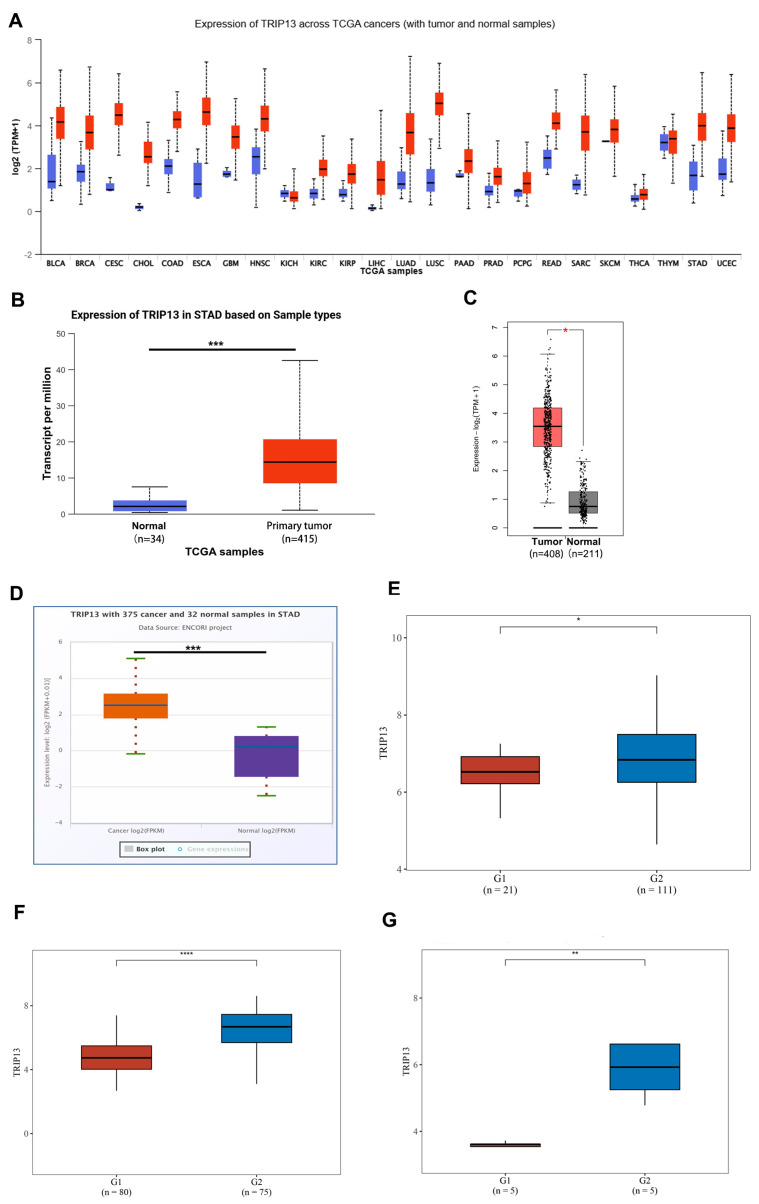
The mRNA expression level of *TRIP13* in GC and healthy tissues. (**A**) The mRNA expression of *TRIP13* in datasets of different cancers compared with normal tissues in the UALCAN database. (**B**,**C**) The mRNA expression of *TRIP13* in GC and normal tissues in the UALCAN (**B**), GEPIA 2.0 (**C**), and ENCORI (**D**) databases. (**E**–**G**) The mRNA levels of *TRIP13* in GC and normal tissues from GSE54129 (**E**), GSE27342 (**F**), and GSE56807 (**G**) databases. * *p* < 0.05; ** *p* < 0.01; *** *p* < 0.001; **** *p* < 0.0001.

**Figure 2 biomedicines-13-02268-f002:**
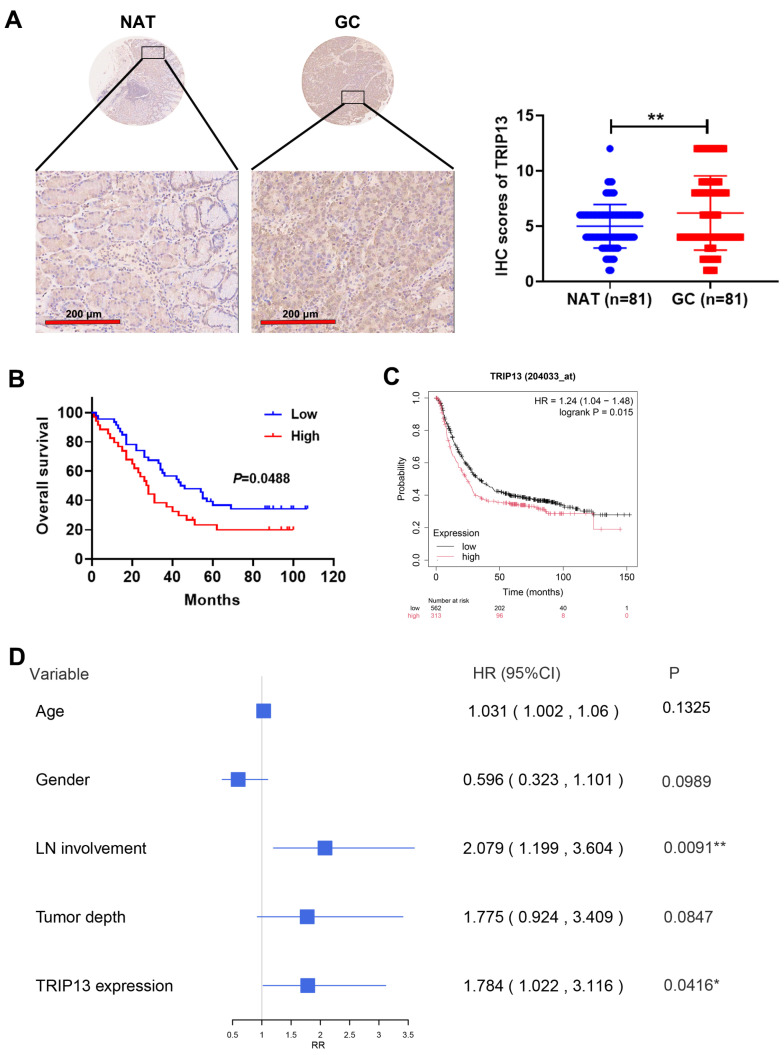
Increased *TRIP13* expression in GC and its association with the inferior prognosis. (**A**) TRIP13 protein levels in gastric cancer (GC) and non-cancerous adjacent tissues (NAT) through immunohistochemistry (IHC). One representative image is shown. Scale bar, 200 μm. (**B**) The overall survival of GC patients was categorized according to high and low TRIP13 expression cohorts, and Kaplan–Meier curves were generated. (**C**) Kaplan–Meier plotter analysis was used to demonstrated the association between TRIP13 expression and prognosis of GC patients. (**D**) Cox regression analysis was conducted to evaluate the association between *TRIP13* and overall survival. HR, Hazardratio. RR, Relarive risk. CI, Confidence interval. * *p* < 0.05, ** *p* < 0.01.

**Figure 3 biomedicines-13-02268-f003:**
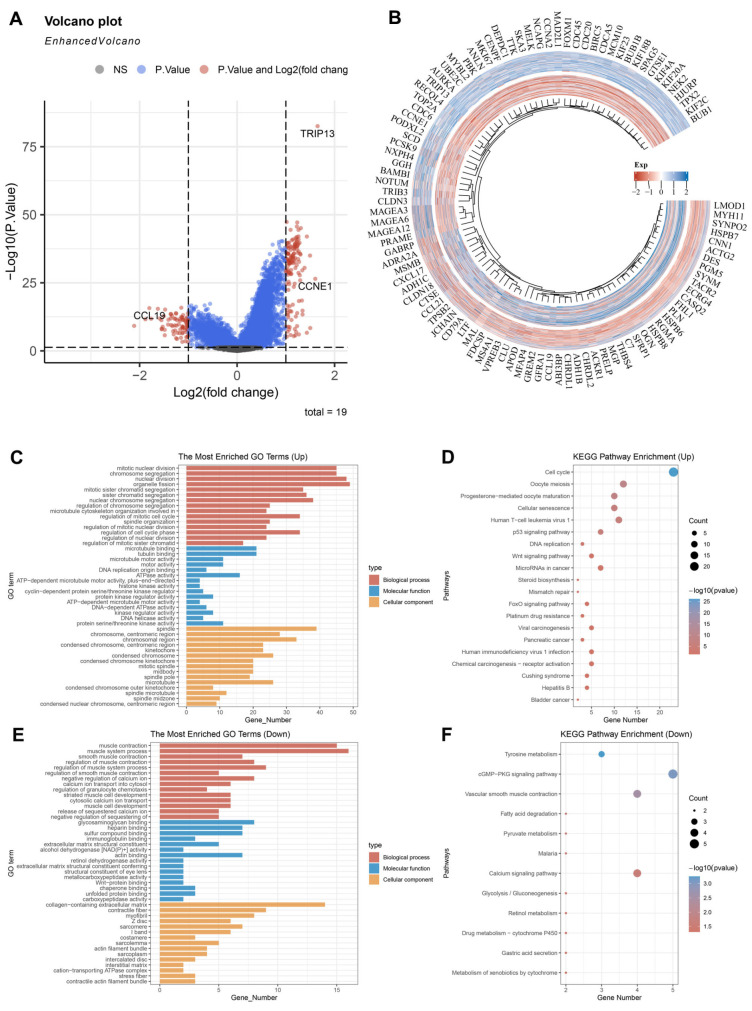
The potential biological role of *TRIP13* in GC. (**A**) A volcano plot illustrates the expression of differentially expressed genes (DEGs) in gastric cancer (GC) tissues with high *TRIP13* expression (right) versus those with low expression (left). (**B**) The heatmap illustrates the fifty most upregulated genes and the fifty most downregulated genes in GC tissues with high *TRIP13* expression compared to those with low expression. (**C**,**D**) The Gene Ontology (GO) (**C**) and Kyoto Encyclopedia of Genes and Genomes (KEGG) (D) pathway enrichment analysis of upregulated DEGs. (**E**,**F**) The GO (**E**) and KEGG (**F**) pathway enrichment analysis of upregulated DEGs.

**Figure 4 biomedicines-13-02268-f004:**
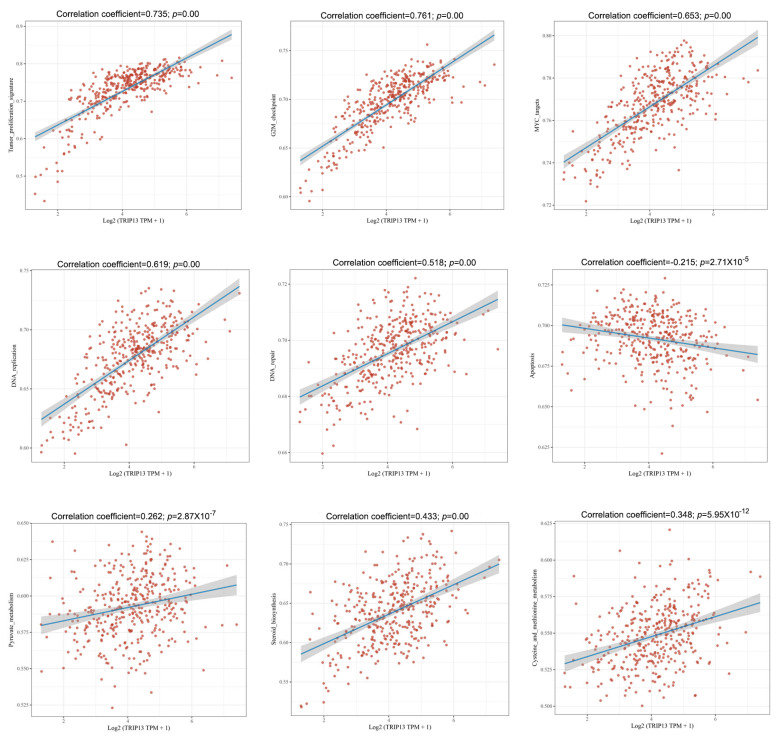
Spearman correlation analysis plot is used to show the correlation between the pathway score and the expression of gene *TRIP13* in GC. G2M, G2 phase and M phase. MYC, MYC proto-oncogene.

**Figure 5 biomedicines-13-02268-f005:**
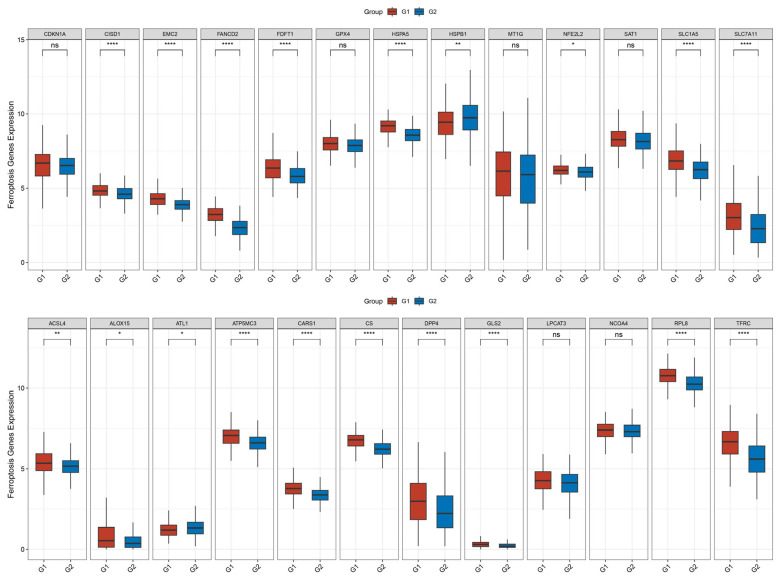
A distribution diagram of the expression of ferroptosis-related genes in in GC tissues with high *TRIP13* expression (G1) versus those with low expression (G2). In this diagram, the horizontal axis represents different ferroptosis-related genes, and the vertical axis represents the expression distribution of these genes. Different colors represent different groups. The number of asterisks represents the level of significance, * *p* < 0.05, ** *p* < 0.01, **** *p* < 0.0001.

**Figure 6 biomedicines-13-02268-f006:**
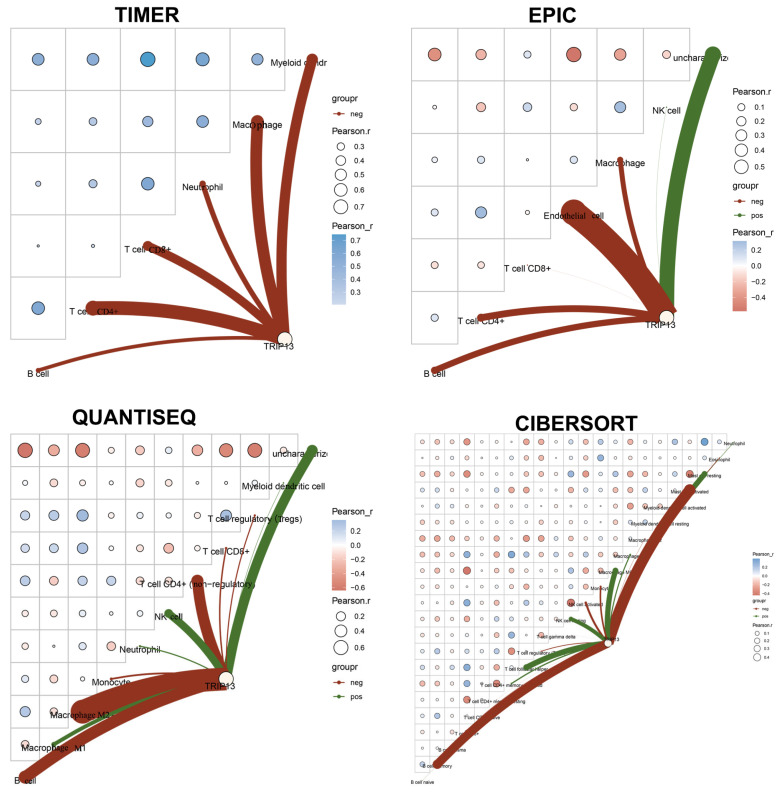
Association between *TRIP13* expression and immunity in gastric cancer (GC). The heatmap illustrates the correlation analysis of *TRIP13* expression with TIMER, EPIC, QUANTISEQ, and CIBERSORT scores, where red represents positive correlation and green represents negative correlation. The redder or greener the color, the greater the correlation between the two; the larger the circle, the stronger the correlation. In the schematic diagram, the red line represents a negative correlation between the model score or gene expression and the immune score, while the green line indicates a positive correlation.

**Figure 7 biomedicines-13-02268-f007:**
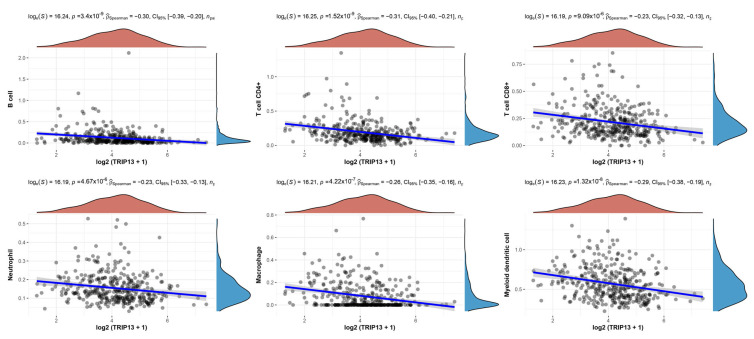
Spearman correlation analysis of *TRIP13* expression with B cells, CD4^+^ T cells, CD8^+^ T cells, macrophages, neutrophils, and myeloid dendritic cells in gastric cancer (GC). CI, Confidence interval.

**Table 1 biomedicines-13-02268-t001:** The association between TRIP13 protein levels and the clinical features of patients with GC.

Characteristic	Total No.	TRIP13 Expression		*p* Value
		Low	High	
All cases	81	46	35	
Age				0.5646
≤65	46	24	22	
>65	35	16	19	
Gender				0.9103
Male	55	31	24	
Female	26	15	11	
LN involvement				0.8884
N0	47	27	20	
N1/N2	34	19	15	
Tumor depth				0.0253
T1/T2	46	8	38	
T3/T4	35	13	22	
AJCC stage				0.1937
IA/IB		4	7	
IIA/IIB		42	28	

## Data Availability

The data of the current study is available from the following open public databases: UALCAN tool (https://ualcan.path.uab.edu/ accessed on 21 April 2025), GEPIA2 (http://gepia2.cancer-pku.cn/#index, accessed on 27 April 2025), and ENCORI (https://rnasysu.com/encori/, accessed on 6 June 2025), as is described above. This study encompassed samples from three databases: GSE54129, which included tumor samples from 111 GC patients post-subtotal gastrectomy and normal gastric mucosa from 21 volunteers during health check-ups, and GSE27342 and GSE56807, which respectively included paired tumor and adjacent normal tissues from 80 and 5 GC patients. Other data will be obtained from the corresponding authors upon reasonable request.

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
