# Peer review of "Decoding TRIP13’s Role in Gastric Cancer: Implications for Prognosis and Immune Response"

_biomedicines, 2025, doi:10.3390/biomedicines13092268_

Round 1

Reviewer 1 Report

Comments and Suggestions for Authors

Shi, et al, report an interesting manuscript of the role of TRIP13 in gastric cancer, including prognosis and immune cells expression. Some points need to be clear.

  1. Please use italics to descript DNA and mRNA.
  2. Line 85, you use auto-select best cutoff function to stratified into low and high expression. What is the software or methods that you used?
  3. High TRIP13 expression is associated with H. pylori related gastric cancer. Please analysis the H. pylori status in TRIP13 expression and prognosis of gastric cancer base on your database.
  4. Figure 6, why do you use TIMP1 expression to immune cell function?
  5. Please mention full name during the first time using abbreviation.
  6. The reference 3 is not complete.

Comments on the Quality of English Language

the english can be improved.

Author Response

Commnets 1: [Please use italics to descript DNA and mRNA.]

Response 1: [We appreciate the suggestion made by the reviewer. We have used italics to descript DNA and mRNA.]

Commnets 2: [Line 85, you use auto-select best cutoff function to stratified into low and high expression. What is the software or methods that you used?]

Response 2: [Thanks. We used the the Kaplan–Meier plotter database (https://kmplot.com/analysis/) to assess the relationship between TRIP13 mRNA levels and overall survival. The Kaplan–Meier plotter database had the auto-select best cutoff function.]

Commnets 3: [High TRIP13 expression is associated with H. pylori related gastric cancer. Please analysis the H. pylori status in TRIP13 expression and prognosis of gastric cancer base on your database.]

Response 3: [According to the reviewer’s suggestion, we employed the UALCAN database to investigate the association between TRIP13 expression and H. pylori-related gastric cancer. As depicted in Supplementary Figure S1, our analysis revealed no significant difference in TRIP13 expression levels between gastric cancer patients with and without H. pylori infection.]

Commnets 4: [Figure 6, why do you use TIMP1 expression to immune cell function?]

Response 4: [We sincerely apologize for the error. After a meticulous review of the data, we identified that the data was mislabeled as TIMP1. We have promptly taken action to correct the error to ensure the accuracy and integrity of our research findings.]

Commnets 5: [Please mention full name during the first time using abbreviation.]

Response 5: [Thanks. We have mentioned full name during the first time using abbreviation.]

Commnets 6: [The reference 3 is not complete.]

Response 6: [Thanks. We have revised the reference 3.]

Commnets 7: [the english can be improved.]

Response 7: [We tried our best to improve the manuscript and made some changes to the manuscript. These changes will not influence the content and framework of the paper. And here we did not list the changes in the revised paper. We appreciate for Editors/Reviewers’ warm work earnestly and hope that the correction will meet with approval.]

Reviewer 2 Report

Comments and Suggestions for Authors

Comments and Suggestions:

Title: Decoding TRIP13's Role in Gastric Cancer: Implications for Prognosis and immune response

Reviewer’s report:

The manuscript by Shi et al., described about the role of TRIP13 expression in different cancer types using databases like UALCAN, GEPIA, GEO, and TIMER. TRIP13 showed elevated expression in Gastric Cancer (GC) patients with IHC showed connection with tumor depth and GO, KEGG pathways highlighted role in cell cycle, DNA repair and regulation of ferroptosis. They concluded that, TRIP13 can be a promising marker of GC.

Although the manuscript does not provide novelty but some points need to be addressed.

  1. Introduction: Please add a small paragraph describing your study in last.
  2. Section 2.1: please include the number of samples taken for analysis in GSE54129, GSE27342, and GSE56807 datasets.
  3. Figure 1A: Add significance levels between tumor and normal samples.
  4. In data availability statement, add information about GEO datasets used (GSE54129, GSE27342, and GSE56807).
  5. Line 113-115: Please add list of ferroptosis associated genes as table form. Also provide details as to how the ferroptosis associated genes were selected as up or downregulated in GC?
  6. Figure 3A: The legends need to be checked and the up and down genes can be colored separately for easy visualization.

Author Response

Commnets 1: [Introduction: Please add a small paragraph describing your study in last.]

Response 1: [We appreciate the insightful comments. We have added a small paragraph describing our study as the final paragraph in the introduction section.]

Commnets 2: [Section 2.1: please include the number of samples taken for analysis in GSE54129, GSE27342, and GSE56807 datasets.]

Response 2: [As suggested, we have included the number of samples taken for analysis in GSE54129, GSE27342, and GSE56807 datasets.]

Commnets 3: [Figure 1A: Add significance levels between tumor and normal samples.]

Response 3: [We appreciate the insightful comments. We have added significance levels between tumor and normal samples.]

Commnets 4: [In data availability statement, add information about GEO datasets used (GSE54129, GSE27342, and GSE56807).]

Response 4: [Thanks. We have added information about GEO datasets used (GSE54129, GSE27342, and GSE56807) in data availability statement.]

Commnets 5: [Line 113-115: Please add list of ferroptosis associated genes as table form. Also provide details as to how the ferroptosis associated genes were selected as up or downregulated in GC?]

Response 5: [This is a nice suggestion. The ferroptosis-related genes are derived from the systematic analysis of the abnormalities and functions of ferroptosis in cancer by Ze-Xian Liu et al (iScience. 2020 Jun 20;23(7):101302.). Then, we utilized the TCGA-STAD dataset to analyze the expression profiles of these genes in GC and adjacent normal tissues. The selection of these genes was based on their established roles in ferroptosis pathways, thereby providing a solid foundation for their classification as upregulated or downregulated in GC. The information was summarized in Table S1.]

Commnets 6: [Figure 3A: The legends need to be checked and the up and down genes can be colored separately for easy visualization.]

Response 6: [This is a great suggestion. We apologize for not being able to distinguish upregulated and downregulated genes using different colors in Figure 3A with our current software. To improve clarity, we have indicated the positions of these genes in the legends of Figure 3A.]